analytical chemistry

extra virgin olive oils, solid-phase microextraction, flavour fingerprint, electronic nose, multivariate analysis

**Author for correspondence:**
Ye Liu
e-mail: liuyecau@126.com

This article has been edited by the Royal Society of Chemistry, including the commissioning, peer review process and editorial aspects up to the point of acceptance.

# Comparison of flavour fingerprint, electronic nose and multivariate analysis for discrimination of extra virgin olive oils

## Qi Zhou[1,2], Shaomin Liu[1], Ye Liu[1] and Huanlu Song[1]

[1]Beijing Engineering and Technology Research Center of Food Additives, Beijing Advanced Innovation Center for Food Nutrition and Human Health, School of Food and Chemical Engineering, Beijing Technology and Business University (BTBU), Beijing 100048, People's Republic of China
[2]Oil Crops Research Institute of the Chinese Academy of Agricultural Sciences, Oil Crops and Lipids Process Technology National & Local Joint Engineering Laboratory, Wuhan, Hubei 430062, People's Republic of China

YL, 0000-0003-1440-4287

Flavour is a special way to discriminate extra virgin olive oils (EVOOs) from other aroma plant oils. In this study, different ratios (5, 10, 15, 20, 30, 50, 70 and 100%) of peanut oil (PO), corn oil (CO) and sunflower seed oil (SO) were discriminated from raw EVOO using flavour fingerprint, electronic nose and multivariate analysis. Fifteen different samples of EVOO were selected to establish the flavour fingerprint based on eight common peaks in solid-phase microextraction–gas chromatography–mass spectrometry corresponding to 4-methyl-2-pentanol, (*E*)-2-hexenal, 1-tridecene, hexyl acetate, (*Z*)-3-hexenyl acetate, (*E*)-2-heptenal, nonanal and α-farnesene. Partial least square discrimination analysis (PLS-DA) was used to differentiate EVOOs and mixed oils containing more than 20% of PO, CO and SO. Furthermore, better discrimination efficiency was observed in PLS-DA than PCA (70% of CO and SO), which was equivalent to the correlation coefficient method of the fingerprint (20% of PO, CO and SO). The electronic nose was able to differentiate oil samples from samples containing 5% mixture. The discrimination method was selected based on the actual requirements of quality control.

## 1. Introduction

Extra virgin olive oil (EVOO) is one of the high-grade edible vegetable oils of the Mediterranean region; it is obtained by cold-pressed extraction from fresh olives (*Olea europaea*) [1]. It can be consumed

in the original form without further industrial refining process, possessing good stability as well as nutritional and health benefits similar to other vegetable oils [2]. Over past decades, the demand for EVOO has been constantly increasing owing to its excellent nutritional and organoleptic properties. In 2014, the production of virgin olive oil was 3.05 million tons throughout the world and increased to 19.27 million tons in 2016 [3].

EVOO's definition originates from its first-class qualities. In the last few years, several techniques have been used for the quality control of EVOO, such as near-, mid- and Fourier-transform infrared spectroscopy [4,5], nuclear magnetic resonance spectroscopy [6], high-performance liquid chromatography–mass spectrometry [7], electro- and sonic-spray ionization mass spectrometry [8], capillary electrophoresis or electrochromatography [9], UV–visible spectroscopy [10] and high-resolution mass spectrometry [11,12].

Aroma is an important criterion for EVOO. Its characteristic flavour is one of the main reasons that distinguishes it from other edible vegetable oils [13]. The aroma of EVOO originates from the autoxidation of unsaturated fatty acid through the lipoxygenase (LOX) pathway [14,15]. The aroma of EVOO is mainly attributed to aldehydes, alcohols, esters, hydrocarbons, ketones and furans [2,15–17]. (E)-2-hexenal, 1-hexanal and (E)-2-heptenal are the important compounds of olive oil identified by various extraction methods such as headspace solid-phase microextraction (SPME), purge and trap (P&T), simultaneous distillation and extraction (SDE) and headspace (HS) [2,16,18,19]. On the other hand, (E)-2-hexenal, (E)-2-hexen-1-ol, (Z)-3-hexen-1-ol and 1-hexanol are found to be the major compounds present in the virgin olive oils obtained from Greece and Tunisia of Koroneiki variety and identified by SPME-gas chromatography (GC)–mass spectrometry (MS) [15].

The identification of key compounds contributing to the aroma is important for the quality evaluation of olive oil. Electronic nose (EN) is an instrument that imitates the sensation of smell and is used for the quality control of olive oils [20,21]. Similarly, multivariate analysis (MA) is often applied to explain the relationship between volatiles and sample types. For example, principal component analysis (PCA) is used to identify linear combinations of volatiles accounting for the amount of data variance and to visualize the trends of sample clusters [17,22,23]. Partial least square discrimination analysis (PLS-DA) is a powerful and extensively used method for the flavour analysis for discriminating samples with different characteristics [23–25].

The fingerprint method has been used as a quality control for different samples. Furthermore, advancement of various analytical methods was able to provide an enormous amount of numerical data about the properties of samples, including chromatography, HPLC, electronic tongue, NMR and isotope and DNA fingerprint [26–32]. The established fingerprint of volatile compounds of bone soup using a combination of GC–MS–olfactometry (O) with PCA provided a reference for category identification [33]. A PLS-DA can establish discrimination criteria to screen the main ionic markers, which is suitable for recognizing and monitoring the quality of flavouring essences [34].

To the best of our knowledge, few studies have reported on the flavour fingerprint (FF) of olive oil. The study 'aroma fingerprint' on EVOO has been focused on the quantitative and qualitative analysis of aroma compounds using DHS or P&T with GC–MS [17,35,36]. And a very recent paper focused on the quality assessment of olive oils based on temperature-ramped HS-GC-IMS and sensory evaluation [37]. Hence, the FF is a feasible attempt to evaluate the quality of EVOOs. For preliminary experiments, SPME (nonsolvent extraction) with better extraction capabilities for more varieties and a higher concentration of compounds in EVOOs was preferred for extracting flavour over solvent-assisted flavour evaporation (SAFE, solvent extraction). The SPME technique can be used to extract volatile compounds rapidly, thus making it more suitable for quality control or authenticity verification. The aim of this study is to compare the classification of FF, EN and MA on the EVOOs according to their aroma components. Fifteen varieties of EVOOs were obtained from Spain, Italy and Greece, and the aroma compounds present in those EVOOs were extracted and analysed using SPME-GC–MS to establish the flavour fingerprint. The EN, FF, PCA and PLS-DA methods were also used to discriminate the samples of EVOOs with mixed oil containing different ratios of peanut oil (PO), corn oil (CO) and sunflower seed oil (SO) (for quality control or authenticity verification of EVOOs, these three kinds of oils with lower price, light colour and flavour were selected).

# 2. Material and methods

## 2.1. Samples

Fifteen varieties of EVOOs from the three largest export countries of olive oil in the world were purchased from the import supermarket. All the samples were authenticated by 'Inspection and

Quarantine Certificate of Entry Goods' from Entry-exit Inspection and Quarantine Bureau of the People's Republic of China. EVOO samples were stored at 15°C in the dark. Specific brands used in this study were as follows: Spain: PL (MUELOLIVA), BDS (BETIS), OL (EURO GOLD), BLN (BELLINA) and YGY (IGAURIN); Italy: OLWL (OLIVOILÀ), OS (OUSA), ALCF (EMROW KITCHEN), MNN (MONINI) and AN (OLITALIA); Greece: XBK (HIPPOCRATES), DMDN (DIAMANDINO), MSWN (MESA VOUNOS), AGL (AEGLE) and DNLE (DONLIAR). All the EVOOs were imported with original packaging from the three countries mentioned above. These samples include the main brands of EVOOs sold in the Chinese market, accounting for more than 95% of consumption. Hence, they have adequate representativeness for the standard of FF.

First-grade squeezed PO of 5S (Luhua), first-grade squeezed CO (Luhua) and SO (Jinlongyu) were purchased from Yonghui supermarket. The mixed samples were prepared using ultrasonic homogenization of a mixture of different oils.

## 2.2. Chemicals

4-methyl-2-pentanol was purchased from Sigma-Aldrich (St Louis, MO, USA). Nitrogen (99.9992% purity) was purchased from Beijing Haipu Beifen Gas Industry Co. Ltd. (Beijing, China).

## 2.3. Aroma extraction of EVOOs using SPME

This was carried out according to the method described by Wang *et al.* with minor modifications [38]. A manual SPME (Supelco Inc., Bellefonte, PA, USA) with a 50/30 µm of divinylbenzene/carboxen/polydimethylsiloxane (DVB/CAR/PDMS) SPME fibre was used for the extraction of volatiles after the fibre was conditioned at 250°C for 30 min. Ten millilitres of EVOO sample was quickly transferred into a 40 ml vial; then 1 µl of 4-methyl-2-pentanol was added as an internal standard at a concentration of 200.5 µg/µl (4-methyl-2-pentanol does not exist in EVOOs and it can be separated from other compounds in the samples). After equilibrium at 60°C for 20 min, the SPME fibre enclosed in a stainless steel needle housing was placed through a hole to expose the fibre at a position of 1 cm above the liquid surface for 40 min. The vials were sealed tightly with screw caps fitted with a Teflon/silicon septum. The vials were continuously swirled during the SPME exposure with an agitation speed of 100 rpm.

## 2.4. Gas chromatography – mass spectrometry analysis

This was carried out following the method reported by Nuzzi *et al.* with minor modifications [39]. The qualitative and quantitative analyses of volatile compounds were performed using an Agilent 7890A gas chromatograph coupled with an Agilent model 7000B series mass spectrometer (GC–MS) and desorbed for 7 min in a split/splitless GC injection port, which was equipped with an inlet linear specific for SPME use (Agilent Technologies, USA). The volatiles were separated on DB-5 and DB-Wax (30 m × 0.25 mm i.d., 0.25 µm, J & W Scientific) silica capillary columns.

The oven temperature was initially set at 40°C, held for 3 min, then increased to 200°C at a rate of 5°C/min, further increased to 230°C at a rate of 10°C/min and held for 3 min and finally held at 250°C for 3 min. The injection port and ionizing source were kept at 250°C and 230°C, respectively. Helium was used as the carrier gas at a flow rate of 1.2 ml min$^{-1}$. The injector mode was splitless. Electron-impact mass spectra were generated at 70 eV, with an *m/z* scan range from 35 to 350 amu. Compounds were identified using NIST 14.0 mass spectra libraries installed in the GC–MS equipment.

## 2.5. FF investigation of EVOOs

The FF of EVOOs was established according to 'Technical Requirements for the Study of Fingerprint of Traditional Chinese Medicine Injection' formulated by the China Food and Drug Administration [40]. The stability of sample, precision of equipment and repeatability of the experiment were investigated before the fingerprint was established. The relative standard deviations (RSDs) of relative retention time ($\alpha$) and relative peak area ($S_r$) of mutual chromatogram peaks were used to establish the fingerprint. In this study, 4-methyl-2-pentanol was used as the reference. The relative retention time ($\alpha$) and relative peak area ($S_r$) were calculated using the following

equations, respectively:

$$\alpha = \frac{t_i}{t_s} \qquad (2.1)$$

and

$$S_r = \frac{A_i}{A_s}, \qquad (2.2)$$

where $\alpha$ is the relative retention time, $t_i$ is the peak time of characteristic peak in the chromatogram, $t_s$ is the peak time of reference peak in the chromatogram, $S_r$ is the relative peak area, $A_i$ is the peak area of characteristic chromatogram peak and $A_s$ is the peak area of reference chromatogram peak.

### 2.5.1. Stability experiments

EVOO samples from the XBK brand were used for the stability experiments. After pretreatment, the samples from different time intervals (2, 4, 6, 8 and 10 h) were analysed by SPME-GC–MS according to the method described in §2.4. The RSD values of relative retention times and relative peak areas were calculated.

### 2.5.2. Precision experiments

EVOO samples from the XBK brand were used for continuous analysis (seven times) using SPME-GC–MS according to the method described in §2.4. The RSD values of relative retention time and relative peak areas were calculated.

### 2.5.3. Repeatability experiment

Fifteen EVOO samples were used for continuous analysis using SPME-GC–MS according to the method described in §2.4. The RSD values of relative retention time and relative peak areas were calculated.

## 2.6. Establishment of FF of EVOOs

Fifteen EVOO samples and samples with different amounts of mixed oils (5, 10, 15, 20, 30, 50, 70 and 100%) of PO, CO and SO were analysed using SPME-GC–MS. The retention time, peak height, peak area and relative peak area of mutual chromatogram peak were calculated. Similarity evaluation system A edition for the chromatographic fingerprint of traditional Chinese medicine was used for the establishment of FF of EVOOs.

## 2.7. Similarity analysis

Included angle cosine (IAC) method, correlation coefficient (CC) method, Euclidean distance and improved Nei coefficient method were used for similarity analysis [41]. The IAC and CC methods were used in this study because of their generality on similarity analysis. The IAC and correlation coefficient values were calculated using the following equations:

$$C_{ir} = \frac{\sum_{k=1}^{m} X_{ik} X_{rk}}{\sqrt{\sum_{k=1}^{m} X^2 ik \sum_{k=1}^{m} X^2 rk}} \qquad (2.3)$$

and

$$r_{ir} = \frac{\sum_{k=1}^{m} (X_{ik} - \overline{X_i})(X_{rk} - \overline{X_r})}{\sqrt{\sum_{k=1}^{m} (X_{ik} - \overline{X_i})^2 \sum_{k=1}^{m} (X_{rk} - \overline{X_r})^2}}, \qquad (2.4)$$

where $C_{ir}$ is the angle cosine value, $r_{ir}$ is the CC value, $X_{ik}$ is the no. k vector of no. i sample, $X_{rk}$ is the no. k vector of standard fingerprint, $\overline{X_i}$ is the mean value of characteristic vectors of no. i sample and $\overline{X_r}$ is the mean value of all the characteristic vectors of standard fingerprint.

## 2.8. EN analysis of EVOOs mixed with other oils

Ten mL oil samples (pure EVOOs and EVOOs with different ratios of PO, CO and SO) were added into a 40 ml headspace bottle of PEN 3 EN (Airsense, Germany). The samples were homogenized with ultrasound. The following conditions were used for EN: 400 s of wash, 10 s of zero adjustment, 5 s of

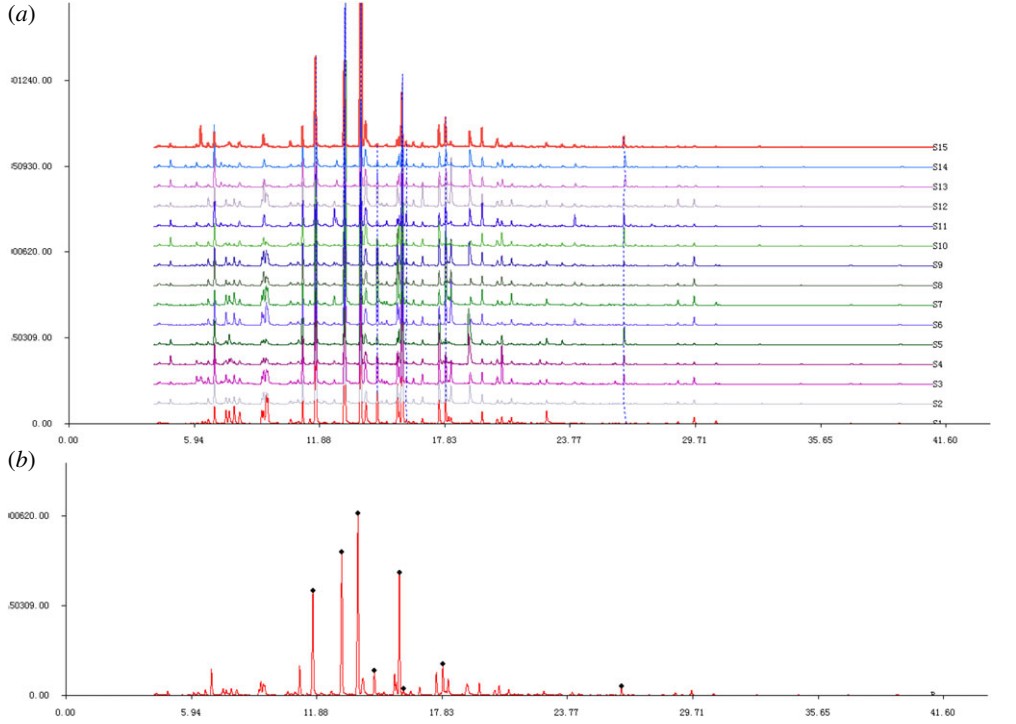

**Figure 1.** (*a*) Common fingerprint of EVOO and (*b*) standard fingerprint of 15 kinds of EVOOs.

sample preparation, 300 s of detection and 300 ml min$^{-1}$ of flow rate for injection. The stable stage of 285–290 s was selected for data collection.

## 2.9. Statistical analysis

PCA and linear discriminant analysis (LDA) of EN data were obtained using the WinMuster software. The PCA of SPME-GC–MS data was carried out using SPSS (Statistical Product and Service Solutions) software, 19th edition. PLS-DA was performed using SIMCA-P+ software, 11th edition.

# 3. Results and discussion

## 3.1. Establishment of FF of EVOOs

GC–MS, a traditional separation method to detect volatile compounds in a complex sample, is also used for the qualitative and quantitative analysis of samples. First, 15 EVOOs were analysed by GC–MS according to the method described in §2. As shown in figure 1*a*, the total ion chromatograms (TICs) of all the 15 EVOOs were similar, because all the samples contain the same common compounds. As is evident from table 1, eight aroma compounds, 4-methyl-2-pentanol, (*E*)-2-hexenal, 1-tridecene, hexyl acetate, (*Z*)-3-hexenyl acetate, (*E*)-2-hepenal, nonanal and α-farnesene, were present in all 15 EVOO samples. All the eight compounds showed characteristic and independent FF. The RSDs of stability, precision and repeatability of peak relative retention time were close to 0, and peak areas were lower than 1 (tables 2–7). Obviously the stability of samples, precision of equipment and repeatability of experimental methods were similar to the fingerprint of the standard. The standard FF of EVOO is shown in figure 1*b*. This method is similar to the study in which 16 and 35 common compounds were used to establish the fingerprints of 10 samples from yak and yellow cattle bone soup [33].

## 3.2. Discrimination evaluation of different mixed oils using FF

Table 8 shows that although the similarities in all 15 EVOOs were observed between IAC and CC methods, some differences in results and sensitivity were still present. For example, the similarity of YGY was lower (0.85), whereas higher similarities of other samples (above 0.90) were observed by

**Table 1.** The common compounds in 15 kinds of EVOOs.

| no. | RT (min) | compounds[a] |
|-----|----------|--------------|
| 1 | 8.96 | 4-methyl-2-pentanol |
| 2 | 9.08 | (E)-2-hexenal |
| 3 | 11.87 | 1-tridecene |
| 4 | 14.73 | hexyl acetate |
| 5 | 14.53 | (Z)-3-hexenyl acetate |
| 6 | 12.85 | (E)-2-hepenal |
| 7 | 17.68 | nonanal |
| 8 | 28.91 | α-farnesene |

RT, retention time.

[a]The compounds were identified using NIST 14.0 mass spectra libraries.

**Table 2.** Stability of peak relative retention time.

| peak no. | 0 h | 2 h | 4 h | 6 h | 8 h | 10 h | RSD |
|----------|-----|-----|-----|-----|-----|------|-----|
| 1 | 1.0000 | 1.0000 | 1.0000 | 1.0000 | 1.0000 | 1.0000 | 0.0000 |
| 2 | 1.1176 | 1.1174 | 1.1189 | 1.1186 | 1.1189 | 1.1174 | 0.0006 |
| 3 | 1.1807 | 1.1802 | 1.1822 | 1.1822 | 1.1828 | 1.1812 | 0.0009 |
| 4 | 1.2481 | 1.2478 | 1.2498 | 1.2497 | 1.2501 | 1.2483 | 0.0008 |
| 5 | 1.3486 | 1.3482 | 1.3502 | 1.3501 | 1.3506 | 1.3490 | 0.0007 |
| 6 | 1.3646 | 1.3645 | 1.3664 | 1.3664 | 1.3671 | 1.3653 | 0.0008 |
| 7 | 1.5246 | 1.5238 | 1.5263 | 1.5264 | 1.5269 | 1.5247 | 0.0008 |
| 8 | 2.2499 | 2.2450 | 2.2482 | 2.2484 | 2.2493 | 2.2461 | 0.0008 |

IAC. Although the similarities of AN, DMDN, DNLE, YGY and XBK ranged from 0.80 to 0.90, those of other samples were above 0.90 by CC. In practice, samples with the smallest similarity values or values below a specific value, for example, 0.85, can be regarded as not acceptable [22]. All the similarities of 15 EVOOs obtained by these two methods were higher than 0.80, thus establishing the FF.

As shown in table 8, the similarity of different mixed oils (PO, CO and SO) was analysed using FF. Only the EVOO samples containing more than 70% of other oils were distinguished by IAC with similarity lower than 0.85. Through correlation analysis, the samples with more than 20% of PO, CO and SO were differentiated with the same similarity standard. The FF showed a good discrimination efficiency on mixed oil samples. Also, a unique mass spectral fingerprint of essence was formed to discriminate the adulterated flavouring essence with a CC ranging from 0.410 to 0.853 [34]. The similarity analysis of yak and yellow cattle bone soup was performed by PCA, and loading capacity was applied to the correlation of samples and volatile compounds [33].

## 3.3. Discrimination evaluation of different mixed oils using EN

As shown in figure 2a, the EVOOs mixed with different amounts of PO were distinguished by PCA. The contribution rate of principal component PC 1 was 97.66%, whereas it was 1.78% for PC 2, and the total contribution rate was 99.44%. This indicates that the selection of the principal component is accurate, and the fitting effect is better. As shown in figure 2c, the contribution rates of PC 1 and PC 2 were 96.51% and 2.42%, respectively, and the total contribution rate was 98.93%. The reproducibility of samples with 5% CO was poor. Furthermore, the discriminating effect of samples with 10% and 15% CO was also unsatisfactory. As shown in figure 2e, the contribution rates of PC 1 and PC 2 were 98.55% and 0.91%, respectively, and the total contribution rate was 99.46%. This shows a good fitting and discriminating effect. However, the reproducibility of samples with 10% CO was poor, and only two parallel samples were used in the analysis. The poor reproducibility of parallel samples was due to the following two reasons: (i) the influence of the external environment on a highly sensitive EN

**Table 3.** Stability of peak relative area.

| peak no. | 0 h | 2 h | 4 h | 6 h | 8 h | 10 h | RSD |
| --- | --- | --- | --- | --- | --- | --- | --- |
| 1 | 1.0000 | 1.0000 | 1.0000 | 1.0000 | 1.0000 | 1.0000 | 0.0000 |
| 2 | 1.7104 | 1.6476 | 2.3896 | 2.3710 | 2.5666 | 1.6747 | 0.2063 |
| 3 | 0.8419 | 0.8674 | 1.4043 | 1.4881 | 1.7045 | 1.0785 | 0.2878 |
| 4 | 0.3778 | 0.3690 | 0.5249 | 0.5105 | 0.5631 | 0.3727 | 0.1968 |
| 5 | 0.6708 | 0.6737 | 0.9825 | 0.9695 | 1.0644 | 0.7154 | 0.2108 |
| 6 | 0.0788 | 0.0844 | 0.1209 | 0.1177 | 0.1313 | 0.0959 | 0.2046 |
| 7 | 0.1911 | 0.2044 | 0.2998 | 0.2992 | 0.3451 | 0.2560 | 0.2257 |
| 8 | 0.0665 | 0.0681 | 0.1765 | 0.2213 | 0.2469 | 0.1292 | 0.5054 |

**Table 4.** Precision of peak relative retention time.

| peak no. | XBK-1 | XBK-2 | XBK-3 | XBK-4 | XBK-5 | XBK-6 | XBK-7 | RSD |
| --- | --- | --- | --- | --- | --- | --- | --- | --- |
| 1 | 1.0000 | 1.0000 | 1.0000 | 1.0000 | 1.0000 | 1.0000 | 1.0000 | 0.0000 |
| 2 | 1.1170 | 1.1177 | 1.1184 | 1.1186 | 1.1180 | 1.1183 | 1.1173 | 0.0005 |
| 3 | 1.1796 | 1.1807 | 1.1819 | 1.1820 | 1.1815 | 1.1824 | 1.1811 | 0.0008 |
| 4 | 1.2468 | 1.2480 | 1.2493 | 1.2495 | 1.2490 | 1.2496 | 1.2482 | 0.0008 |
| 5 | 1.3471 | 1.3486 | 1.3500 | 1.3502 | 1.3496 | 1.3500 | 1.3486 | 0.0008 |
| 6 | 1.3630 | 1.3646 | 1.3663 | 1.3665 | 1.3660 | 1.3664 | 1.3650 | 0.0009 |
| 7 | 1.5225 | 1.5241 | 1.5261 | 1.5262 | 1.5255 | 1.5263 | 1.5247 | 0.0009 |
| 8 | 2.2440 | 2.2458 | 2.2483 | 2.2484 | 2.2475 | 2.2495 | 2.2467 | 0.0008 |

**Table 5.** Precision of peak relative area.

| peak no. | XBK-1 | XBK-2 | XBK-3 | XBK-4 | XBK-5 | XBK-6 | XBK-7 | RSD |
| --- | --- | --- | --- | --- | --- | --- | --- | --- |
| 1 | 1.0000 | 1.0000 | 1.0000 | 1.0000 | 1.0000 | 1.0000 | 1.0000 | 0.0000 |
| 2 | 1.5225 | 1.7953 | 2.2083 | 2.1927 | 2.0464 | 2.1063 | 1.7255 | 0.1356 |
| 3 | 0.8361 | 1.0204 | 1.2912 | 1.3701 | 1.2170 | 1.4906 | 1.1846 | 0.1820 |
| 4 | 0.3605 | 0.3825 | 0.4535 | 0.4852 | 0.4433 | 0.4640 | 0.3885 | 0.1119 |
| 5 | 0.6526 | 0.7145 | 0.8602 | 0.9286 | 0.8410 | 0.8833 | 0.7383 | 0.1262 |
| 6 | 0.0772 | 0.0909 | 0.1113 | 0.1210 | 0.1077 | 0.1144 | 0.0958 | 0.1495 |
| 7 | 0.1848 | 0.2216 | 0.2687 | 0.2933 | 0.2654 | 0.3018 | 0.2491 | 0.1604 |
| 8 | 0.0785 | 0.1033 | 0.1774 | 0.1859 | 0.1372 | 0.2947 | 0.2130 | 0.4271 |

signal; and (ii) the defect in the algorithm of PCA because it is an unsupervised algorithm [42]. Through PCA, the entire group of data was mapped to coordinate axes that can express them easily. Hence, the data classification information was not used during this process. This not only reduced the dimensionality but also weakened the clustering effect due to the increase in variance.

As shown in figure 2b, the EVOOs mixed with a different ratio of PO were distinguished by LDA. The contribution rates of PC 1 and PC 2 were 66.38% and 18.26%, respectively, and the total contribution rate was 84.64%. This indicates that these two principal components provided most of the information on PO containing EVOOs. As shown in figure 2d,f, the contribution rates of PC 1 were 64.05% and 55.29%, and those of PC 2 were 24.59% and 30.90%. The total contribution rates were 88.64% and 86.19%, respectively. This also shows good fitting and discriminating effect on the CO and SO containing EVOOs. Notably, LDA discriminated different mixed oils using EN with good clustering effect and reproducibility of

**Table 6.** Repeatability of peak relative retention time.

| peak no. | AGL | ALCF | AN | BDS | BLN | DMDN | DNLE | MSWN | MNN | OLWL | OL | OS | PL | XBK | YGY | RSD |
|---|---|---|---|---|---|---|---|---|---|---|---|---|---|---|---|---|
| 1 | 1.0000 | 1.0000 | 1.0000 | 1.0000 | 1.0000 | 1.0000 | 1.0000 | 1.0000 | 1.0000 | 1.0000 | 1.0000 | 1.0000 | 1.0000 | 1.0000 | 1.0000 | 0.0000 |
| 2 | 1.1169 | 1.1157 | 1.1173 | 1.1145 | 1.1135 | 1.1170 | 1.1212 | 1.1154 | 1.1166 | 1.1148 | 1.1123 | 1.1169 | 1.1135 | 1.1176 | 1.1146 | 0.0019 |
| 3 | 1.1800 | 1.1777 | 1.1793 | 1.1802 | 1.1806 | 1.1791 | 1.1814 | 1.1814 | 1.1795 | 1.1813 | 1.1787 | 1.1794 | 1.1804 | 1.1801 | 1.1837 | 0.0012 |
| 4 | 1.2467 | 1.2455 | 1.2467 | 1.2472 | 1.2466 | 1.2474 | 1.2484 | 1.2475 | 1.2468 | 1.2470 | 1.2454 | 1.2470 | 1.2469 | 1.2474 | 1.2466 | 0.0006 |
| 5 | 1.3490 | 1.3461 | 1.3482 | 1.3484 | 1.3481 | 1.3484 | 1.3504 | 1.3489 | 1.3482 | 1.3483 | 1.3459 | 1.3481 | 1.3474 | 1.3479 | 1.3471 | 0.0008 |
| 6 | 1.3638 | 1.3622 | 1.3638 | 1.3646 | 1.3642 | 1.3642 | 1.3655 | 1.3646 | 1.3641 | 1.3646 | 1.3624 | 1.3640 | 1.3640 | 1.3639 | 1.3637 | 0.0006 |
| 7 | 1.5227 | 1.5219 | 1.5232 | 1.5242 | 1.5233 | 1.5238 | 1.5252 | 1.5244 | 1.5234 | 1.5237 | 1.5220 | 1.5236 | 1.5240 | 1.5234 | 1.5238 | 0.0006 |
| 8 | 2.2417 | 2.2413 | 2.2431 | 2.2455 | 2.2436 | 2.2441 | 2.2446 | 2.2438 | 2.2433 | 2.2444 | 2.2423 | 2.2433 | 2.2478 | 2.2440 | 2.2436 | 0.0007 |

**Table 7.** Repeatability of peak relative area.

| peak no. | AGL | ALCF | AN | BDS | BLN | DMDN | DNLE | MSWN | MNN | OLWL | OL | OS | PL | XBK | YGY | RSD |
|---|---|---|---|---|---|---|---|---|---|---|---|---|---|---|---|---|
| 1 | 1.0000 | 1.0000 | 1.0000 | 1.0000 | 1.0000 | 1.0000 | 1.0000 | 1.0000 | 1.0000 | 1.0000 | 1.0000 | 1.0000 | 1.0000 | 1.0000 | 1.0000 | 0.0000 |
| 2 | 1.5372 | 1.1379 | 1.6799 | 0.4018 | 0.2825 | 1.4115 | 3.5365 | 0.8145 | 1.3838 | 0.6769 | 0.1216 | 1.4775 | 0.1291 | 1.6717 | 0.6371 | 0.7735 |
| 3 | 0.8732 | 0.4830 | 0.7181 | 1.0175 | 1.2483 | 0.4887 | 1.1655 | 1.2552 | 0.8284 | 1.4196 | 0.6950 | 0.7995 | 0.9188 | 0.9492 | 2.5183 | 0.4823 |
| 4 | 0.2651 | 0.0988 | 0.2207 | 0.0762 | 0.0520 | 0.2552 | 0.5135 | 0.3102 | 0.2098 | 0.0797 | 0.0378 | 0.2428 | 0.0430 | 0.4039 | 0.0355 | 0.7720 |
| 5 | 1.3353 | 0.4139 | 0.9422 | 0.8306 | 0.8977 | 0.8497 | 1.6508 | 0.9366 | 0.9520 | 0.7387 | 0.2457 | 0.8094 | 0.3010 | 0.7345 | 0.3452 | 0.4738 |
| 6 | 0.0346 | 0.0371 | 0.0551 | 0.0324 | 0.0183 | 0.0564 | 0.0633 | 0.0636 | 0.0624 | 0.0602 | 0.0619 | 0.0660 | 0.0385 | 0.0823 | 0.0416 | 0.3279 |
| 7 | 0.1477 | 0.1294 | 0.1895 | 0.0788 | 0.0365 | 0.2023 | 0.5420 | 0.3507 | 0.1888 | 0.1050 | 0.0997 | 0.1962 | 0.0964 | 0.2011 | 0.1564 | 0.6841 |
| 8 | 0.0068 | 0.0087 | 0.0570 | 0.0373 | 0.0313 | 0.0082 | 0.0094 | 0.0088 | 0.0121 | 0.0255 | 0.0196 | 0.0137 | 0.0219 | 0.0994 | 0.0210 | 0.9695 |

**Table 8.** Similarity analysis of 15 kinds of EVOOs and different mixed oils.

| method | sample | | | | | | | | | | | | | | |
|---|---|---|---|---|---|---|---|---|---|---|---|---|---|---|---|
| | AGL | ALCF | AN | BDS | BLN | DMDN | DNLE | MSWN | MNN | OLWL | OL | OS | PL | YGY | XBK |
| included angle cosine | 0.96 | 0.98 | 0.94 | 0.96 | 0.95 | 0.91 | 0.92 | 0.99 | 0.97 | 0.95 | 0.96 | 0.97 | 0.96 | 0.85 | 0.93 |
| correlation coefficient | 0.93 | 0.97 | 0.89 | 0.93 | 0.92 | 0.84 | 0.87 | 0.98 | 0.96 | 0.93 | 0.90 | 0.90 | 0.90 | 0.81 | 0.88 |

| method | sample | | | | | | | | | | | | | | |
|---|---|---|---|---|---|---|---|---|---|---|---|---|---|---|---|
| | 20% PO | 30% PO | 50% PO | 70% PO | 100% PO | 20% CO | 30% CO | 50% CO | 70% CO | 100% CO | 20% SO | 30% SO | 50% SO | 70% SO | 100% SO |
| included angle cosine | 0.93 | 0.92 | 0.84 | 0.73 | 0.33 | 0.92 | 0.91 | 0.87 | 0.77 | 0.37 | 0.92 | 0.92 | 0.90 | 0.75 | 0.37 |
| correlation coefficient | 0.88 | 0.87 | 0.71 | 0.47 | −0.13 | 0.82 | 0.85 | 0.77 | 0.64 | 0.23 | 0.87 | 0.86 | 0.82 | 0.62 | 0.24 |

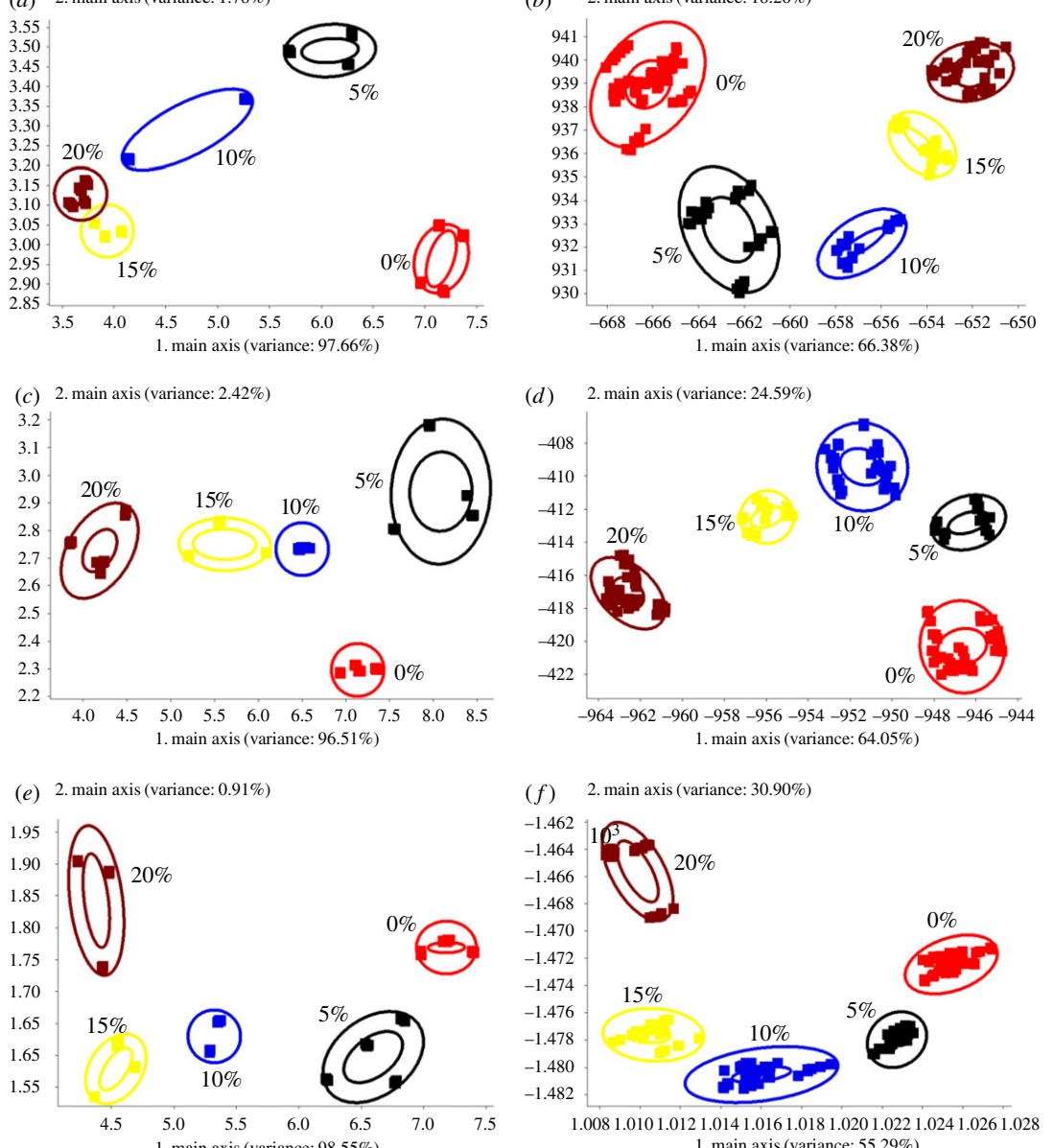

**Figure 2.** PCA and LDA of EVOOs with peanut oil, corn oil and sunflower seed oil at different ratios. (*a*) PCA of peanut oil; (*b*) LDA of peanut oil; (*c*) PCA of corn oil; (*d*) LDA of corn oil; (*e*) PCA of sunflower seed oil; (*f*) LDA of sunflower seed oil.

parallel samples. Unlike PCA, LDA is a supervised algorithm [42]. The optimized coordinate axis was used while reducing dimensionality due to the data with the label. It can classify the datapoints within the category and differentiate the datapoints between different categories.

## 3.4. Discrimination evaluation of different mixed oils using MA (PCA and PLS-DA)

PCA was performed based on the concentration of 11 key aroma compounds obtained from EVOOs and mixed oils, and a PCA model was constructed (figure 3). The key aroma compounds are (*E*)-2-hexenal, 1-tetradecene, 2-methylpyrazine, hexyl acetate, (*Z*)-3-hexenyl acetate, (*E*)-2-heptenal, 2,5-dimethylpyrazine, nonanal, 3,5-dimethyl-2-ethyl pyrazine, furfural and α-farnesene. As shown in figure 3*a*, the sample points of all the EVOOs and those containing below 50% of SO and CO are concentrated in the Y area. This indicates that PCA differentiated the mixed oil samples containing more than 20% of PO. However, lower discrimination efficiencies were observed for the samples with more than 70% of CO and SO. As is evident from figure 3*a,b*, (*E*)-2-hexenal, (*Z*)-3-hexenyl acetate, hexyl acetate and nonanal were the most potent compounds of the characteristic aroma components

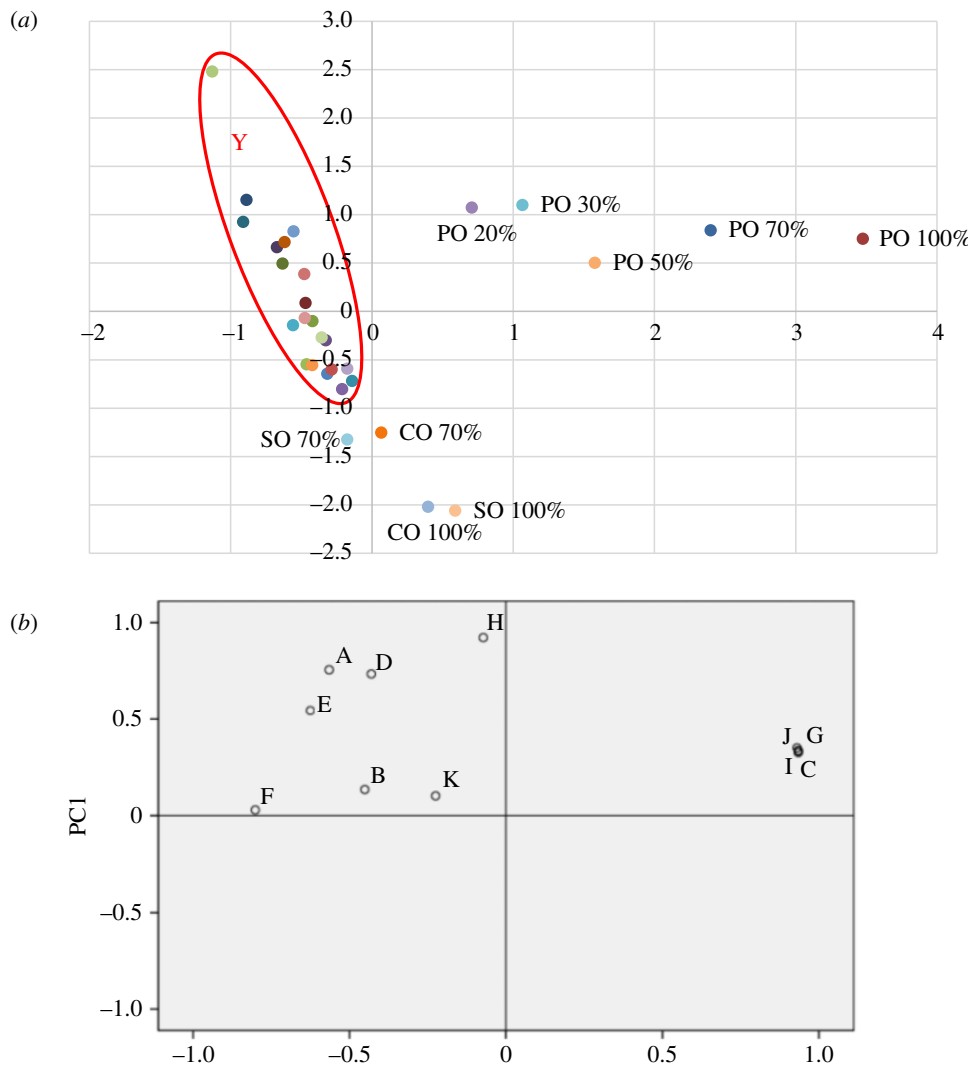

**Figure 3.** Score (*a*) and loading (*b*) plot of PCA of EVOOs and mixed oils. (In loading plot, (*a*) (*E*)-2-hexenal, *b*: 1-tetradecene, (*c*) 2-methylpyrazine, (*d*) hexyl acetate, (*e*) (*Z*)-3-hexenyl acetate, (*f*) (*E*)-2-hepenal, (*g*) 2,5-dimethylpyrazine, (*h*) nonanal, (*i*): 3,5-dimethyl-2-ethyl pyrazine, (*j*) furfural, (*k*) α-farenene).

in EVOOs. The secondary components were (*E*)-2-heptenal, 1-tetradecene and α-farnesene. On the other hand, 2-methylpyrazine, 2,5-dimethylpyrazine, 3,5-dimethyl-2-ethyl pyrazine and furfural were the most potent compounds in PO. However, no characteristic aroma components were present in CO and SO due to the extremely low flavour. The results are consistent with flavour analysis and EN. Compared to the FF, PCA had a similar effect on the discrimination of different oil samples. The results are consistent with the differences between tobacco leaf samples and flavour samples obtained by PCA [22]. The similarities were performed on the fingerprint data of 11 Shexiang and 36 Cablin samples using PCA [43,44]. The samples at different distillation stages of the same origin, especially JNC liquors, were located close to each other, and they were not satisfactorily discriminated by this approach [23].

A PLS-DA model was conducted to discriminate EVOOs mixed with different oils according to the data matrix of their volatile compounds. As shown in figure 4*a*, EVOOs and mixed oils with more than 20% of PO, CO and SO were differentiated by PLS-DA. However, the distance between mixed oils containing CO and SO was close due to the similar and light flavour of CO and SO. The discrimination efficiency of PLS-DA was better than that of PCA and FF. Figure 4*b* shows that DA1 representing EVOOs existed in the fourth quadrant, DA2 representing EVOOs containing PO existed in the third quadrant and DA3 and DA4 representing EVOOs containing CO and SO, respectively, existed in the first quadrant. This distribution is consistent with the result shown in figure 3*b*.

(a)

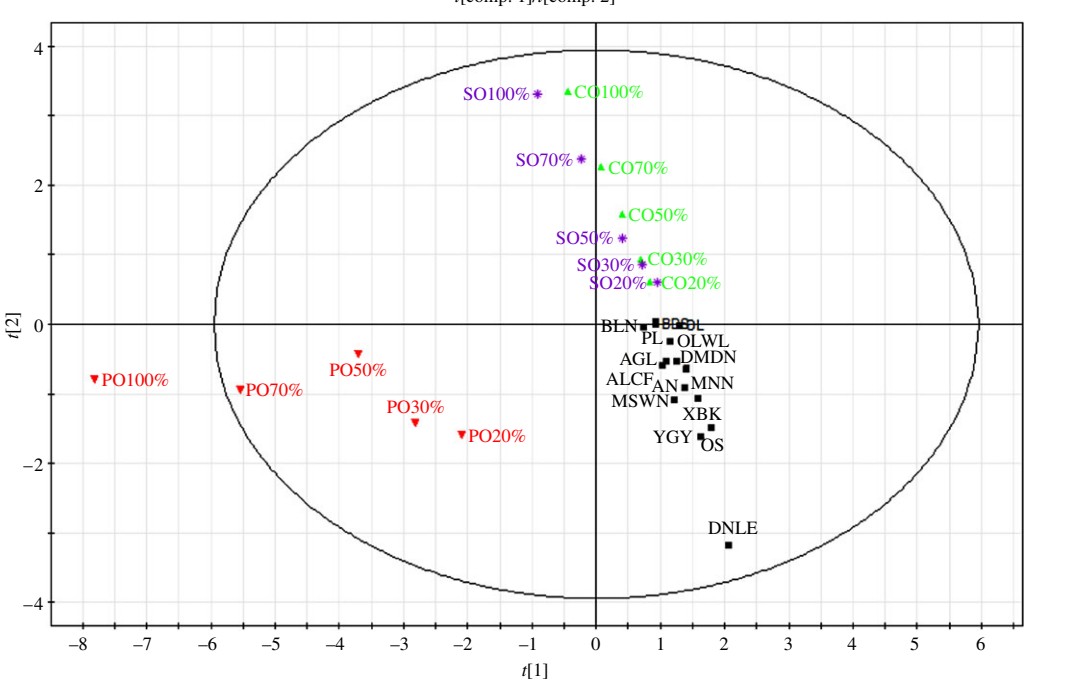

R2X[1] = 0.476321    R2X[2] = 0.228314    ellipse: hotelling T2 (0.95)

(b)

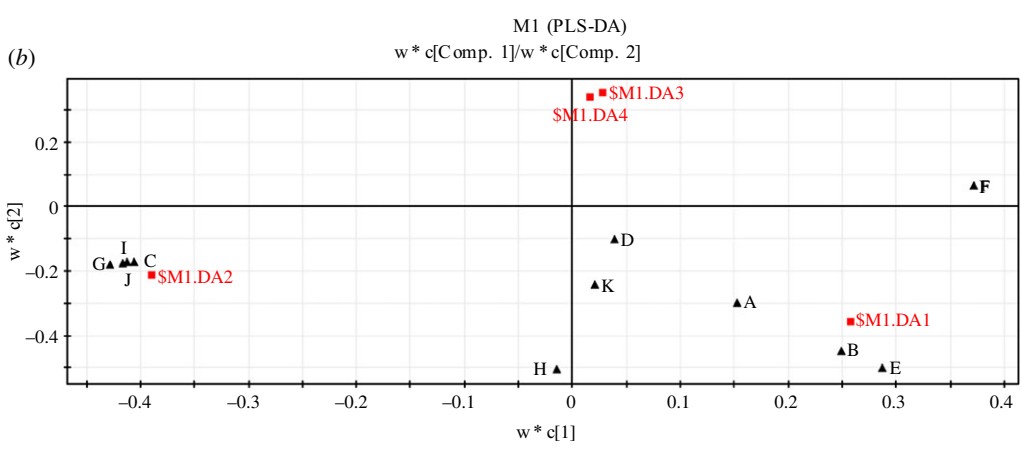

R2X[1] = 0.476321    R2X[2] = 0.228314

**Figure 4.** Score (*a*) and loading (*b*) plot of PLS-DA of EVOOs and mixed oils. In the loading plot, (A) (*E*)-2-hexenal, (B) 1-tetradecene, (C) 2-methylpyrazine, (D) hexyl acetate, (E) (*Z*)-3-hexenyl acetate, (F) (*E*)-2-hepenal, (G) 2,5-dimethylpyrazine, (H) nonanal, (I) 3,5-dimethyl-2-ethyl pyrazine, (J) furfural and (K) α-farenene.

1-Tetradecene, (*Z*)-3-hexenyl acetate, (*E*)-2-hexenal, hexyl acetate and α-farnesene were scattered in the fourth quadrant, indicating that they were the most potent compounds in EVOOs. By contrast, 2-methylpyrazine, 2,5-dimethylpyrazine, 3,5-dimethyl-2-ethyl pyrazine and furfural approached DA2, suggesting that they had a great contribution to the flavour of EVOOs containing PO. Only (*E*)-2-heptenal existed in the first quadrant, and it was far from DA3 and DA4. This is due to the low flavour of CO and SO, and they diluted the flavour of EVOOs when added. This result is consistent with that of PCA. Similarly, the mid-level data fusion approach using PLS-DA scores was found to be the best strategy for defective versus nondefective olive oil discrimination by headspace-mass spectrometry, mid-infrared spectroscopy and UV–visible spectroscopy [25]. Based on the quantification results of 21 relevant aroma active compounds, PLS-DA was performed to classify the quality of olive oils with the variables and predicted 88% correctly [24]. PLS-DA not only provided powerful ability to discriminate different raw liquors in relation with the deviation of concentrations of volatile compounds but also depicted the volatile markers of each type of raw liquors, especially the heart distilling stage of raw liquors [23].

## 3.5. Comparison of discrimination effect of FF, EN and MA on different mixed oils

As is evident from the above results, EN can differentiate the oil samples with 5% mixture, and the best discrimination ratio of FF and MA was 20%. The precision of EN was better than FF and MA. However, the sensitivity of EN was extremely high, and the results were affected by a slight change in external condition. Thus, many duplicate experiments were required to reduce the error, which was time-consuming. EN also suffers from a drawback that it cannot determine the flavour compounds of samples comprehensively due to the selectivity of sensors. All these factors limit the application of EN to the quality control of EVOOs. While FF and MA were based on the SPME method, which have good stability and repeatability, they can complete the extraction and analysis rapidly. The SPME can also investigate the difference between EVOOs and mixed oils. In actual condition, the adulteration ratio of EVOOs was always higher than 20%. Hence, FF and MA can also satisfy the requirements for efficient detection and discrimination.

## 4. Conclusion

Eight compounds, including 4-methyl-2-pentanol, (E)-2-hexenal, 1-tridecene, hexyl acetate, (Z)-3-hexenyl acetate, (E)-2-hepenal, nonanal and α-farnesene were identified in all 15 EVOOs used, and all of them have characteristic FF. The similarities of all the 15 EVOOs were higher than 0.80 based on IAC and CC, thus establishing the FF. The samples containing more than 20% of PO and CO were well differentiated. In other words, FF showed an excellent discrimination between mixed oil samples. Similarly, the LDA of EN also discriminated different mixed oils. Good clustering effect and reproducibility of parallel samples were observed. It can classify the datapoints within the category and differentiate the datapoints between different categories. EVOOs and mixed oils containing more than 20% of PO, CO and SO were well differentiated by PLS-DA. Among the three methods, EN showed the best discrimination effect on mixed oil samples. For the actual condition, the discrimination effects of FF and MA were also competent for the quality control of EVOOs.

Data accessibility. All data are included in the article.
Authors' contributions. Q.Z. collected samples, drafted and revised the manuscript; S.L. carried out the laboratory work and performed statistical analysis; Y.L. conceived, designed and coordinated the study; H.S. helped to edit and revise the manuscript. All authors gave final approval for publication.
Competing interests. We declare we have no competing interests.
Funding. Financial support was provided by The National Key Research and Development Program of China (no. 2016YFD0401104), National Natural Science Foundation of China (no. 31871838) and Support Project of High-level Teachers in Beijing Municipal Universities (no. IDHT20180506).
Acknowledgements. The authors gratefully acknowledge the financial support from the National Natural Science Foundation of China and Ministry of Science and Technology of China.

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
