## [Reviewer comments · Royal Society Open Science]

Review History

RSOS-190002.R0 (Original submission)

Review form: Reviewer 1

Is the manuscript scientifically sound in its present form?

Yes

Are the interpretations and conclusions justified by the results?

Yes

Is the language acceptable?

Yes

Is it clear how to access all supporting data?

Yes

Do you have any ethical concerns with this paper?

No

Have you any concerns about statistical analyses in this paper?

No

Recommendation?

Accept with minor revision (please list in comments)

Comments to the Author(s)

The work titled as "Comparison of flavor fingerprint, electronic nose, and multivariate analysis for discrimination of extra virgin olive oils " has been turned into an interesting way of presenting the data. Considering that the results of the study may be useful for subsequent studies, it may be accepted after the revisions mentioned below.

1-Line 62-63 : In introduction; the authors must be clear about their findings. Avoid the use of suspicious statements using "might be".

2-Line 70-73: Please provide 3-4 references of such studies to strengthen the statement.

3-Line 90-91: please note the brands of EVOOs from three countries.

4-Line 100: delete "(CR)"

5-Line 109:What is I.S.? Internal standard, I assume. Then, I suggest rephrasing as "...pentanol was added as an internal standard at the concentration of...";

6-Table 1:the footnote (b) should be in according with the state of Line 129-130.

7 -Line 170-171: Whether some references could be provided to support the statement.

8 -Line 205-207: Please add some significance levels. Result presentation in abstract should be quantitative.

Review form: Reviewer 2

Is the manuscript scientifically sound in its present form?

Yes

Are the interpretations and conclusions justified by the results?

Yes

Is the language acceptable?

Yes

Is it clear how to access all supporting data?

Yes

Do you have any ethical concerns with this paper?

No

Have you any concerns about statistical analyses in this paper?

No

Recommendation?

Accept with minor revision (please list in comments)

Comments to the Author(s)

The manuscript describes in quite a clear way to compare flavor fingerprint, electronic nose, and multivariate analysis for discrimination of extra virgin olive oils. The research is original and the information is needed for the field of quality control of olive oils. Nevertheless, there are some

detail problems in the manuscript. I suggest that it can be considered for publication after minor revisions.

Specific comments:

1. Line 96-98: Why were peanut oil, corn oil, and sunflower seed oil selected to prepare the mixed oil system? It should be explained.
2. Line 109: Why was 4-methyl-2-pentanol selected as the internal standard? It should be explained.
3. Line 126: Change "...250 °C and 230 °C..." to "...250 and 230 °C...".
4. Line 132-133: "Technical Requirements for the Study of Fingerprint of Traditional Chinese Medicine Injection" should be listed in the "References" section.
5. Line 162: It is better to change "adulteration oils" to "mixed oils".
6. Line 167-168: Delete the sentence "The number of reference chromatogram was set as S1".

Decision letter (RSOS-190002.R0)

06-Feb-2019

Dear Dr Liu:

Title: Comparison of flavor fingerprint, electronic nose, and multivariate analysis for discrimination of extra virgin olive oils
 Manuscript ID: RSOS-190002

Thank you for submitting the above manuscript to Royal Society Open Science. On behalf of the Editors and the Royal Society of Chemistry, I am pleased to inform you that your manuscript will be accepted for publication in Royal Society Open Science subject to minor revision in accordance with the referee suggestions. Please find the reviewers' comments at the end of this email.

The reviewers and handling editors have recommended publication, but also suggest some minor revisions to your manuscript. Therefore, I invite you to respond to the comments and revise your manuscript.

Please also include the following statements alongside the other end statements. As we cannot publish your manuscript without these end statements included, if you feel that a given heading is not relevant to your paper, please nevertheless include the heading and explicitly state that it is not relevant to your work. We have included a screenshot example of the end statements for reference.

- Ethics statement

Please clarify whether you received ethical approval from a local ethics committee to carry out your study. If so please include details of this, including the name of the committee that gave consent in a Research Ethics section after your main text. Please also clarify whether you received informed consent for the participants to participate in the study and state this in your Research Ethics section.

OR

Please clarify whether you obtained the necessary licences and approvals from your institutional animal ethics committee before conducting your research. Please provide details of these licences and approvals in an Animal Ethics section after your main text.

OR

Please clarify whether you obtained the appropriate permissions and licences to conduct the fieldwork detailed in your study. Please provide details of these in your methods section.

Because the schedule for publication is very tight, it is a condition of publication that you submit the revised version of your manuscript before 15-Feb-2019. Please note that the revision deadline will expire at 00.00am on this date. If you do not think you will be able to meet this date please let me know immediately.

Best wishes,
Dr Laura Smith
Publishing Editor, Journals

Royal Society of Chemistry
Thomas Graham House

Science Park, Milton Road
Cambridge, CB4 0WF
Royal Society Open Science - Chemistry Editorial Office

RSC Associate Editor:
Comments to the Author:
(There are no comments.)

RSC Subject Editor:
Comments to the Author:
(There are no comments.)

Reviewer comments to Author:
Reviewer: 1

Comments to the Author(s)

The work titled as "Comparison of flavor fingerprint, electronic nose, and multivariate analysis for discrimination of extra virgin olive oils " has been turned into an interesting way of presenting the data. Considering that the results of the study may be useful for subsequent studies, it may be accepted after the revisions mentioned below.

1-Line 62-63 : In introduction; the authors must be clear about their findings. Avoid the use of suspicious statements using "might be".

2-Line 70-73: Please provide 3-4 references of such studies to strengthen the statement.

3-Line 90-91: please note the brands of EVOOs from three countries.

4-Line 100: delete "(CR)"

5-Line 109:What is I.S.? Internal standard, I assume. Then, I suggest rephrasing as "...pentanol was added as an internal standard at the concentration of...";

6-Table 1:the footnote (b) should be in according with the state of Line 129-130.

7 -Line 170-171: Whether some references could be provided to support the statement.

8 -Line 205-207: Please add some significance levels. Result presentation in abstract should be quantitative.

Reviewer: 2

Comments to the Author(s)

The manuscript describes in quite a clear way to compare flavor fingerprint, electronic nose, and multivariate analysis for discrimination of extra virgin olive oils. The research is original and the information is needed for the field of quality control of olive oils. Nevertheless, there are some detail problems in the manuscript. I suggest that it can be considered for publication after minor revisions.

Specific comments:

1. Line 96-98: Why were peanut oil, corn oil, and sunflower seed oil selected to prepare the mixed oil system? It should be explained.

2. Line 109: Why was 4-methyl-2-pentanol selected as the internal standard? It should be explained.

3. Line 126: Change "...250 °C and 230 °C..." to "...250 and 230 °C...".

4. Line 132-133: "Technical Requirements for the Study of Fingerprint of Traditional Chinese Medicine Injection" should be listed in the "References" section.
5. Line 162: It is better to change "adulteration oils" to "mixed oils".
6. Line 167-168: Delete the sentence "The number of reference chromatogram was set as S1".

Author's Response to Decision Letter for (RSOS-190002.R0)

See Appendix A.

Decision letter (RSOS-190002.R1)

19-Feb-2019

Dear Dr Liu:

Title: Comparison of flavor fingerprint, electronic nose, and multivariate analysis for discrimination of extra virgin olive oils

Manuscript ID: RSOS-190002.R1

It is a pleasure to accept your manuscript in its current form for publication in Royal Society Open Science. The chemistry content of Royal Society Open Science is published in collaboration with the Royal Society of Chemistry.

RSC Associate Editor
Comments to the Author:
(There are no comments.)

Reviewer(s)' Comments to Author:

Appendix A

Dear Editor:

Thank you for your comments concerning this manuscript entitled **“Comparison of flavor fingerprint, electronic nose, and multivariate analysis for discrimination of extra virgin olive oils”** (Manuscript ID: RSOS-190002). Those comments are all valuable and very helpful for improving this manuscript. We have studied comments carefully and have made correction which we hope meet with approval. The main corrections in the paper and the responds to technical check results are as flowing:

Responds to the referees' comments:

Referee #1

1. Line 62-63: In introduction; the authors must be clear about their findings. Avoid the use of suspicious statements using "might be".

Responds: Thanks for your suggestion! The “might be” was changed to “is” and marked in red.

2-Line 70-73: Please provide 3-4 references of such studies to strengthen the statement.

Responds: Thanks for your suggestion! However, there was no report on the application of SAFE on flavor compound extract of EVOOs previously. This statement was based on the preliminary experiments in our lab, and the results were not published so far. Hence, there is no relevant reference to be cited. And there might be some misunderstanding on the expression. So it is more definite that “in EVOOs” was added in this sentence and marked in red.

3-Line 90-91: please note the brands of EVOOs from three countries.

Responds: Thanks for your suggestion! The brands of EVOOs from three countries were added and marked in red.

4-Line 100: delete “(CR)”

Responds: Thanks for your suggestion! The “(CR)” was deleted.

5-Line 109: What is I.S.? Internal standard, I assume. Then, I suggest rephrasing as “...pentanol was added as an internal standard at the concentration of...”

Responds: Thanks for your suggestion! The “I.S.” was deleted, then the expression was changed to “1 μL of 4-methyl-2-pentanol was added as an internal standard at a concentration of 200.5 $\mu\text{g}/\mu\text{L}$ ” and marked in red.

6-Table 1:the footnote (b) should be in according with the state of Line 129-130.

Responds: Thanks for your suggestion! The footnote (b) in Table 1 was changed according to the state of Line 129-130 and marked in red.

7 –Line 170-171: Whether some references could be provided to support the statement.

Responds: Thanks for your suggestion! A reference ([41]) was added to support the statement and marked in red.

8 –Line 205-207: Please add some significance levels. Result presentation in abstract should be quantitative.

Responds: Thanks for your suggestion! The RSDs were only applied to test if the stability of samples, precision of equipment, and repeatability of experimental satisfied the requirement of fingerprint establishment. It did not concern the significance analysis, and there was no significance level.

In addition, the discrimination efficiency of PCA and correlation coefficient method of fingerprint were added in the section “Abstract” quantitatively and marked in red.

Referee #2

1. Line 96-98: Why were peanut oil, corn oil, and sunflower seed oil selected to prepare the mixed oil system? It should be explained.

Responds: Thanks for your suggestion! For quality control or authenticity verification of EVOOs, the oils with lower price, light color and flavor were selected. These three kinds of oils satisfy requirements above. So they were purchased for the experiment samples in this study. It was illustrated in the last section of “1. Introduction” and marked in red.

2. Line 109: Why was 4-methyl-2-pentanol selected as the internal standard? It should be explained.

Responds: Thanks for your suggestion! Based on the blank experiment, 4-methyl-2-pentanol did not exist in EVOO samples in this study. And 4-methyl-2-pentanol could be separated from other compounds in the samples. So it was taken as the internal standard. The illustration was added in the section “2.3. Aroma extraction of EVOOs using SPME” and marked in red.

3. Line 126: Change “...250 °C and 230 °C...” to “...250 and 230 °C...”.

Responds: Thanks for your suggestion! The expression was changed to “...250 and 230 °C...” and marked in red.

4. Line 132-133: “Technical Requirements for the Study of Fingerprint of Traditional Chinese Medicine Injection” should be listed in the “References” section.

Responds: Thanks for your suggestion! This “Technical Requirements” was listed in the section “References [40]” and marked in red.

5. Line 162: It is better to change “adulteration oils” to “mixed oils”.

Responds: Thanks for your suggestion! The expression “adulteration oils” was changed to “mixed oils” and marked in red.

6. Line 167-168: Delete the sentence “The number of reference chromatogram was set as S1”.

Responds: Thanks for your suggestion! The sentence “The number of reference chromatogram was set as S1” was deleted.